# Diversity of Leaf Fungal Endophytes from Two *Coffea arabica* Varieties and Antagonism towards Coffee Leaf Rust

**DOI:** 10.3390/plants13060814

**Published:** 2024-03-12

**Authors:** Ruth A. Poma-Angamarca, Jacqueline R. Rojas, Aminael Sánchez-Rodríguez, Mario X. Ruiz-González

**Affiliations:** 1Departamento de Ciencias Biológicas y Agropecuarias, Universidad Técnica Particular de Loja, San Cayetano Alto s/n, Loja 1101608, Ecuador; ruthalex2003@yahoo.es (R.A.P.-A.); jerojasx@utpl.edu.ec (J.R.R.); asanchez2@utpl.edu.ec (A.S.-R.); 2SENESCYT is the Secretaría de Educación Superior, Ciencia, Tecnología e Innovación from the Government of Ecuador, Proyecto Prometeo SENESCYT, Universidad Técnica Particular de Loja, San Cayetano Alto s/n, Loja 1101608, Ecuador

**Keywords:** biocontrol, biodiversity, Caturra, Colombia, *Colletotrichum* sp., *Hemileia vastatrix*, *Xylaria* sp.

## Abstract

Coffee has immense value as a worldwide-appreciated commodity. However, its production faces the effects of climate change and the spread of severe diseases such as coffee leaf rust (CLR). The exploration of fungal endophytes associated with *Coffea* sp. has already found the existence of nearly 600 fungal species, but their role in the plants remains practically unknown. We have researched the diversity of leaf fungal endophytes in two *Coffea arabica* varieties: one susceptible and one resistant to CLR. Then, we conducted cross-infection essays with four common endophyte species (three *Colletotrichum* sp. and *Xylaria* sp. 1) and *Hemileia vastatrix* (CLR) in leaf discs, to investigate the interaction of the endophytes on CLR colonisation success and severity of infection. Two *Colletotrichum* sp., when inoculated 72 h before *H. vastatrix*, prevented the colonisation of the leaf disc by the latter. Moreover, the presence of endophytes prior to the arrival of *H. vastatrix* ameliorated the severity of CLR. Our work highlights both the importance of characterising the hidden biodiversity of endophytes and investigating their potential roles in the plant-endophyte interaction.

## 1. Introduction

Coffee is one of the most precious commodities of day-to-day life. Most of the coffee production (mainly *Coffea arabica* and *C. canephora*) occurs in developing countries in Africa, America, and Asia. Moreover, the economies of up to 25 million small producers rely exclusively on coffee cultivation and successful harvests (e.g., [1]). In addition to both the volatility of the coffee market and agricultural input prices, there exist, however, two main biophysical concerns about coffee productivity: (i) the instability and unpredictability of the weather conditions due to climate change (e.g., drought, extreme temperatures, heavy rains); and (ii) the spread of crop pests and diseases, sometimes facilitated by the new environmental conditions (e.g., berry borer, leaf rust). Ecuador produces around 8000 t of coffee in 30,000 ha (SIPA, http://sipa.agricultura.gob.ec/, accessed on 21 February 2024), which represents an annual income of about USD 150 million (http://www.camae.org/, accessed on 21 February 2024).

The effects of climate change on coffee production might have a negative impact on both the producers and the international market [2]. In fact, coffee is very sensitive to variations in both temperatures and rainfall [3], with an optimal temperature range for Arabica coffee production between 18–22 °C (e.g., [4,5]. Under up-to-date climatic predictions, we expect a drastic reduction of 19–50% of the suitable areas for coffee production [3,6,7]. Increased local temperatures might exert a stronger negative impact on coffee plantations in the Andes and in the Amazon. Moreover, locally, climate change predictions for Ecuador suggest both a temperature increase by 2–3 °C and a rainfall increase of 3% by 2050 (https://climateknowledgeportal.worldbank.org/country/ecuador/climate-data-projections, accessed on 5 December 2022). The latter translates into a potential loss of up to 20% of the suitable area for Arabica coffee cultivation and a rise in the range of elevation suitable for cultivation to 1000–2800 m a.s.l. [6,8].

A major risk of increasing temperatures is the potential spread of coffee leaf rust (CLR), the most important disease affecting *C. arabica* worldwide, because at temperatures above 23 °C the pathogenic fungus can exhibit up to 2000 times higher sporulation than at the optimal coffee temperature range [9]. CLR is caused by the obligate biotrophic basidiomycete fungus *Hemileia vastatrix* Berk. et Br. (Pucciniomycetes, Pucciniales, Zaghouaniaceae). This disease, which spread all over the world from Ceylon during the last 150 years, has disastrous effects on production, with up to 40% losses in Latin America (e.g., the 2008 to present outbreak), and up to 70% in Asia as an indirect consequence of defoliation; that is, economic losses of US$ 1–2 billion annually [10,11,12,13,14,15]. CLR affects green leaves and prospers better in humid and shady conditions [16]. CLR epidemiology, however, is complex and seems to depend mainly on plantation characteristics and altitude, where wind, rain, and animals play their role in the dispersion of uredospores [17,18]. To date, the preferred control measures are the use of fungicides, the replacement of old susceptible cultivars with resistant ones, and agroecological control [14,15].

The use of fungal endophytes for either the improvement of crops or the biocontrol of plant pathogens and pests is a reality [19]. Fungal endophytes reside within the living tissues of all plants as mutualists or commensals without causing evidence of disease during all or part of their lives [20]. The advantages of using wild endophytes compared to traditional agrochemicals as a tool to enhance plant health and protection against harsh conditions, pathogens, or pests, are the reduction of both the environmental impact and crop production costs while improving soil health (e.g., [21,22,23,24]. Endophytes are asymptomatic microbes that invade, at some stage, different plant tissues (e.g., [25]. Indeed, the presence of endophytes inside plant tissues confers advantages to the host plant [26] and represents a potential alternative to chemical pesticides [27] with beneficial effects for species of agricultural importance. Furthermore, endophytes may become an important tool to boost plant tolerance and resilience against climate change effects [28]. Previous works on coffee found bacterial endophytes acting as plant growth promoters, or potential agents to control bacterial or fungal diseases, such as CLR [29,30,31]. Moreover, some endophytic fungi behave as entomopathogens or mycoparasites of plant fungal diseases, such as *Colletotrichum gloeosporoides* or *Lecanicillium lecanii* against CLR in coffee [32,33,34].

Coffee plants exhibit high levels of microbial endophyte diversity in Latin America, but their antagonistic effects against CLR remain practically unknown [31,34,35,36,37,38,39]. In America, the endophyte genera with the greatest geographic distribution in coffee plants are: *Colletotrichum* sp., *Fusarium* sp., and *Xylaria* sp. [37,38,39,40,41]

In this work, we investigated the fungal endophyte community associated with two common *C. arabica* varieties, Caturra and Colombia, from southern Ecuador (Shucos, Loja province). The former, being susceptible, and the latter, being resistant to CLR, provide an opportunity to explore the potential interaction between the host genotype and the fungal endophyte selection in the field. Then, to evaluate the hypothesis that some of these asymptomatic fungal species might provide some protection to the host, we selected four common and widespread coffee endophyte species [37,38,39,40,41]: two *Colletotrichum* species that are known pathogens of different host plant species, *C. lupini* and *C. karstii* [42,43], an undescribed species from the *Colletotrichum acutatum* species complex, and a fungus from a genus known both to prosper on decaying plant material and produce a wide variety of bioactive secondary metabolites, *Xylaria* sp. 1 [44,45]. With these four fungi, we tested their potential antagonistic effects against *H. vastatrix* in the Caturra susceptible cultivar by investigating colonisation success and the severity of *H. vastatrix* in coffee leaf discs.

## 2. Results

### 2.1. Sampling, Isolation, and Endophyte Identification

We isolated 114 endophytes from 44 potentially different species (Table 1) present either in Caturra (14), Colombia (24), or both (6) varieties. The endophytic fungi took between three and twelve days to grow from the plant tissue after sowing the leaf fragments (Appendix A). The closest taxa to 31 of the isolated fungal morphotypes in this work have been described as endophytes, and to eight as pathogens (Appendix A). Only two fungal morphotypes have been previously found in *Coffea arabica*, a *Colletotrichum* sp. from Puerto Rico and *Xylaria curta* from Colombia. Six endophyte taxa were previously isolated from Ecuador, one *Annulohypoxylon* sp. and five *Xylaria* species, the latter with percentages of sequence identity lower than 97%. The relative frequencies of endophyte morphospecies found in the Caturra variety were quite low, up to 25%. However, in the Colombia variety we found three highly prevalent fungi: *Colletotrichum karstii*, *Xylaria* sp. 1, and *Xylaria* sp. 7, with a relative frequency of 87.50%. *Xylaria* sp. 1 was also present in Caturra (12.50% relative frequency).

The values near zero of the unbiased Simpson’s dominance indexes in Caturra and Colombia mean that we have many different species, more or less equally represented within each *Coffea* variety, without a dominant taxon over others (Table 2). Thus, we found very high levels of biodiversity. The bias-corrected Shannon entropy *H_cs_* value in Caturra implies that the diversity might be higher in this variety compared to the estimate in the Colombia variety due to the presence of many rare OTUs and potentially undiscovered species. However, both *H_cs_* estimates are above three, thus highlighting the presence of many species evenly represented. The Margalef’s diversity index, which assesses species richness while compensating for the effects of sample size, points to Colombia as the variety with the highest richness of endophyte species, in terms of number of species. The two beta-diversity indexes produced high values of dissimilarity when comparing the endophyte communities present in Caturra and Colombia, thus suggesting that both communities have a different composition of species. The PERMANOVA analysis, which found significant differences between the endophyte community compositions of Caturra and Colombia (*F*_1,13_ = 2.4628; *p*-value = 0.0026), confirmed the latter observation.

### 2.2. Cross-Inoculation Essays in Leaf Discs

#### 2.2.1. Colonisation Success

We found that the treatment had significant effects on the colonisation success of leaf discs by CLR when exposed to all the endophytes except *C. karstii* (Table 3 and Figure 1 and Figure 2). Post hoc analysis only found significant differences among treatments for CLR in interaction with *Colletotrichum* sp. 1 (adjusted *z* = 3.16, *p*-value = 0.00158; Figure 1). Overall, it is observed that when the endophyte has been previously inoculated, the CLR colonisation success is reduced (Figure 2). The latter was confirmed when analysing for differences within each treatment for *Colletotrichum* sp. 1 (*χ*^2^ = 13.333, d.f. = 1, *p*-value < 0.001) and *C. lupini* (*χ*^2^ = 18.027, d.f. = 1, *p*-value < 0.001). When inoculating *C. kartstii* and CLR at the same time, the colonisation of the endophyte was significantly higher than that of CLR (*χ*^2^ = 6.144, d.f. = 1, *p*-value = 0.013).

#### 2.2.2. CLR Severity in Leaf Discs Infection

Overall significant effects of treatment on *H. vastatrix* severity were found when using the Kruskal-Wallis test (Table 4). Inoculating the endophyte 72 h before de CLR produced the lowest severity compared to coinfection at the same time (post-hoc after Bonferroni correction: *p*-value < 0.001), 72 h after the CLR (*p*-value = 0.042) or the CLR alone (*p*-value < 0.001). Each endophyte species had significant treatment effects on CLR severity (Table 4). Thus, we found pairwise significant differences in severity among treatments for *Colletotrichum* sp. 1, *C. karstii* and *Xylaria* sp. 1 (Figure 3).

## 3. Discussion

We found differences in the composition of the fungal endophyte communities associated with two varieties of *C. arabica*. The resistant variety, Colombia, exhibited higher levels of richness and higher densities of endophytes than the susceptible variety, Caturra. However, the unbiased Shannon index suggests that more endophyte diversity is expected to be found in the Caturra variety than in Colombia. The genetic analysis of the fungi strongly suggests that many of our endophytes are new species to science, and the description of the closest taxa supports their ecological endophytic mode of life. Moreover, the inoculation of leaf discs of Caturra coffee with an endophyte prior to CLR inoculation has a protective effect by ameliorating CLR colonisation and severity. However, although we used 10-month-old young plants that were raised in a greenhouse, it is possible that their leaves were already colonised with unidentified endophytes, and that could have contributed to ameliorating CLR severity. Nevertheless, our results are exploratory, and more experiments should be conducted on living plants instead of on leaf discs before determining the practical applicability of these endophytic fungi.

The most diverse taxonomic groups in our analysis are *Colletotrichum* spp. (15 genotypes) and the Xylariaceae (10 genotypes), as found in previous works on Latin American coffee trees. Overall, and along with the taxonomic groups herein described, we found in the literature 593 fungal endophytic species belonging to 183 different taxonomic groups associated with aerial parts of *C. arabica* from Latin America, Africa, Asia, and Oceania [29,36,37,38,39,40,41,46,47]. Thirty-six of the taxonomic groups found in our work exhibited less than 98.5% of identity with known fungal species, a standard cut-off for species identification (e.g., [48]). That means that they might represent new species. This high degree of diversity is not surprising, as recent work highlights that the diversity of coffee endophytes is driven by both coffee genotype and geographical characteristics [49]. The two coffee varieties here investigated coexist in the same geographical area under similar climatic conditions and crop management; thus, the difference in the composition of their endophyte communities may be due mainly to host variety and endophyte genotype-genotype interactions, as well as endophyte colonisation success. Furthermore, six of our taxa have not been isolated outside of Ecuador and have not been previously reported in coffee: *Annulohypoxylon* sp., *Hypoxylon* sp., *Lopadostoma* sp., *Musicillium* sp., *Preussia* sp., and one Cladosporiaceae. An unknown Cladosporiaceae, one *Penicillium* spp., several *Colletotrichum* spp., and the entomopathogenic *Beauveria* sp., could potentially behave as mycoparasites or serious antagonists of *H. vastatrix*, as previously reported [29,36,50].

The inoculation of leaf discs with *Colletotrichum* sp. 1 and *C. lupini* endophytes before the inoculation of the pathogen strongly reduced the colonisation success of the pathogen; and the presence of *C. karstii* or *Xylaria* sp. 1, strongly decreased the CLR severity in the susceptible variety. Overall, the inoculation of endophyte species prior to the arrival of the pathogen decreased the severity of the disease in leaf discs. Thus, our findings strongly support previous research stating the importance of both the order of arrival of the endophyte and the pathogen and infection to promote resistance [51]. In addition, although a *Colletotrichum* sp. strain has been found to improve growth and plant secondary metabolism [52], the genus hosts many pathogenic fungi, and, thus, the inoculation of plants with these species must be carefully monitored under different environmental situations and host genetic backgrounds, as well as to avoid breakthrough infections in other crop hosts. Xylariaceae members, however, produce a great diversity of secondary metabolites, and some species even show antagonistic effects against pathogens [44,53], thus representing a source of unknown bioactive components with potential application in agronomy, industry, medicine, and biotechnology. Indeed, focusing on studying endophyte isolates not only contributes to characterizing biodiversity and microbial specificity, but also represents the opportunity to bio-prospect new compounds with great potential for the development of green technologies and promoting sustainable practices.

While the differences in susceptibility to *H. vastatrix* between Colombia and Caturra have a strong genetic component [54], the fungal endophyte community might be playing an important role in host defence through direct interactions with the pathogens [21]. In fact, we observe that both colonisation success and severity were much lower when any endophyte species was inoculated first. The output of biotic interactions, however, is complex, and changes in the environment might drive changes in the lifestyle, from mutualist to saprophyte or pathogenic [55].

Inoculating leaves with local endophyte propagules could prevent or reduce the drastic effects of *H. vastatrix*. If applied, this new management strategy may help to reduce the economic and environmental impact of using chemical pesticides and promote the conservation of coffee varieties that otherwise would be replaced with more resistant ones. The endophytes from the Colombia variety successfully grew in Caturra variety leaf discs, indicating their adaptability across different host genotypes. However, it appears that there is a fungal preference for specific host genotypes in the field. New experiments should first investigate the potential pathogenicity of these endophytes in whole coffee plants before exploring their protective role against different CLR strains.

## 4. Materials and Methods

### 4.1. Sampling, Isolation, and Endophyte Identification

We sampled three phenotypically healthy leaves from eight plants of two varieties of *C. arabica*, Caturra and Colombia, at two sites from Shucos (Loja) between September and November 2016, at an altitude of 1987 m a.s.l. (03°55′55.5″ S, 079°13′17.7” W) and 2003 m a.s.l. (03°56′27.6” S, 079°13′08.3” W), respectively. Both sites correspond to the low dry montane forest (bs-MB) formation [56] and the Aw Köpper-Geiger climate. The weather conditions exhibited an average annual minimum/maximum temperature of 12.9/22.6 °C, precipitation of 780 mm, and 83% relative humidity. The owners identified the coffee varieties that were not confirmed genotypically. The leaves were stored in aseptic paper bags and transported to the laboratory within an hour.

Fungal endophytes were isolated after the Arnold et al. [57] protocol. Leaves were processed within 48 h post-harvesting. For each leaf, we obtained 48 segments (1 × 2 mm), which were surface sterilised for two minutes in a 0.525% NaOCl solution, two minutes in 70% EtOH, and a final wash in distilled water. We cultured the segments in malt yeast dextrose agar (MYDA) at lab temperature (20 °C) for 60 days. Overall, we cultured 1152 segments for each coffee variety. Each segment was monitored daily under the stereoscopic microscope to check for the presence of fungal hyphae. According to the definition of endophyte as asymptomatic fungi (or bacteria) found inside the plant tissues [58], we only considered as endophytes those fungi that grew after a minimum of three days of culturing, and whose hyphae came directly from the inner part of the tissue. Then, we transferred them to new MYDA plates under sterile conditions and incubated them for 2–3 weeks at 25 °C. DNA was extracted from pure fresh mycelia using the Chelex^®^ method [59], and the ITS region was amplified using universal primers ITS1F and ITS4 [60,61]. PCR products were sequenced by Macrogen (Seoul, Republic of Korea, https://dna.macrogen.com, accessed on 5 May 2017). Raw sequences were edited in Chromas 2.6.5 (Technelysium, South Brisban, QLD, Australia, http://technelysium.com.au/wp/chromas/, accessed on 7 March 2024), and compared to the NCBI nucleotide databases using blastn (https://blast.ncbi.nlm.nih.gov/Blast.cgi, accessed on 20 December 2023) to identify taxonomic groups.

### 4.2. Endophyte Biodiversity

We calculated in Past 4.09 [62] three standard alpha biodiversity indexes, the taxonomic diversity and distinctness to evaluate the taxonomic relatedness of both communities of endophytes, and two beta diversity indexes to compare the diversity of endophytes, and the similarity between both endophyte communities in each coffee variety: (1) unbiased Simpson’s Index of Dominance: $D=\frac{\Sigma_{i=1}n_{i}(n_{i}-1)}{N(N-1)}$, which provides a measure of taxon dominance; (2) the Chao & Shen [63] bias-corrected Shannon entropy index (*H_CS_*), which takes into account missing species, sample coverage, and the relative abundances of species in the sample; and (3) Margalef Richness Index:$\;K=\frac{S-1}{lnN}$, where *S* is the total number of species and *N* is the total number of individuals. The latter is a species diversity index that is easy to interpret and complements very well the Simpson’s dominance and the bias-corrected Shannon’s entropy indexes. We did not calculate abundance-based coverage estimators such as Chao1 or ACE because many endophytes are singleton or doubleton species. Finally, we calculated the beta biodiversity index of Whitaker and the $\beta_{sim}=\frac{\min\left(b,c\right)}{\min\left(b,c\right)+a}$ after Koleff et al. [64].

### 4.3. Cross-Inoculation Assays in Leaf Discs

Fungal endophytes might provide protection against major pathogens [21]. Therefore, we performed a preliminary analysis of the interaction of four endophyte species to test their potential antagonistic effects against CLR infection. The choice of the endophytes was based on the following complementary criteria: (1) species already found in previous studies [37,38,39,65]; (2) the ecological role of the species (plant pathogen or saprophyte) and the symbiotic lifestyle; (3) endophyte occurrence in Caturra or Colombia (e.g., in both varieties, or specific fungi from the CLR resistant Colombia variety); and (4) the biology and behaviour of the endophyte (high sporulation ability, fast or slow growers). Thus, we chose two *Colletotrichum* taxa from the *acutatum* complex and one *C. karstii* that were present only in the Colombia variety, and one *Xylaria* sp. 1 endophyte, present in both the Caturra and Colombia varieties. The *Colletotrichum* species have been demonstrated to exhibit different symbiotic lifestyles, depending on their host species [66]. *Hemileia vastatrix* spores were brush collected from infected field Caturra leaves.

We set underside up 2.0 cm diameter discs, cut out of healthy leaves from 10 month-old Caturra plants, on soaked paper and randomly assigned 20 discs to each treatment and three controls: (i) ‘E + R72′, endophyte (E) 72 h before inoculating CLR (R); (ii) ‘E + R0′, both endophyte and CLR inoculated simultaneously at time 0; (iii) ‘R + E72′, endophyte inoculated 72 h after CLR; (iv) ‘E’, the endophyte alone; (v) ‘R’, CLR alone; and (vi) ‘_d_H_2_O’, blank control. We inoculated the discs by putting 100 µL of _d_H_2_O containing 1 × 10^5^ propagules for each endophyte and *H. vastatrix*, on the surface of the leaf discs; and incubated with a 12 h photoperiod at 22 ± 2 °C and an RH of 100%, after Silva et al. [31]. Thus, we maximised the potential adhesion of spores to the leaf surface and the finding of stomata for successful infection [54]. CLR symptoms in the leaf discs were like those of a standard infection, showing from small pale-yellow spots to bigger spots with masses of urediniospores (Figure 1). However, we did not perform any histological preparation of the discs to evaluate the mode of penetration.

Twenty-five days post-inoculation leaf discs were evaluated for the presence of endophyte and CLR symptoms using the software ASSESS 2.0 (L. Lamari, American Phytopathological Society, St. Paul, MN, USA). The colonisation success of both endophytes and *H. vastatrix* was determined by the presence of each fungal species growing on the surface of the leaf discs. We recorded the severities using a 0 to 5 scale [31], based on the percentage of leaf surface damaged (0: 0%; 1: 0–2.5%; 2: 2.5–5%; 3: 5–15%; 4: 15–25%, and 5: >25%). To compare the results, a severity percentage of 100 was assigned to the severity observed in the control treatment; then, the severity observed in the other treatments was compared with the control and expressed as a severity index (DSI) according to [31].

### 4.4. Statistical Analysis

We used a nonparametric PERMANOVA analysis with 9999 permutations (RVAideMemoire [adonis] v 0.9-73 in vegan: Community Ecology Package) on endophyte taxonomic groups (presence/absence data) at the plant level to test for differences in the composition of endophyte-associated communities between both coffee varieties, Caturra and Colombia.

We analysed the effects of the treatments on the colonisation success of the endophytes and *H. vastatrix* with a three-way cross-tab Pearson’s Chi-square test, with colonisation success as the response variable, treatment as the explanatory variable, and organism (endophyte or CLR) as the control variable. We remove from the statistical analysis the endophyte and CLR alone inoculation treatments, as well as the H_2_O control, after assessing that no contamination has occurred in either of them, to evaluate the effects of the interaction. Chi-square post-hoc tests were conducted on the adjusted standardised residuals. Then, we performed a Chi-square test for each treatment to analyse for differences between the colonisation frequencies of each endophyte and the CLR.

We tested whether there were differences among CLR *severities* for the different inoculation treatments using a Kruskal-Wallis test in IBM SPSS Statistics v24 and post-hoc pairwise comparisons corrected for significance after Bonferroni.

## Figures and Tables

**Figure 1 plants-13-00814-f001:**
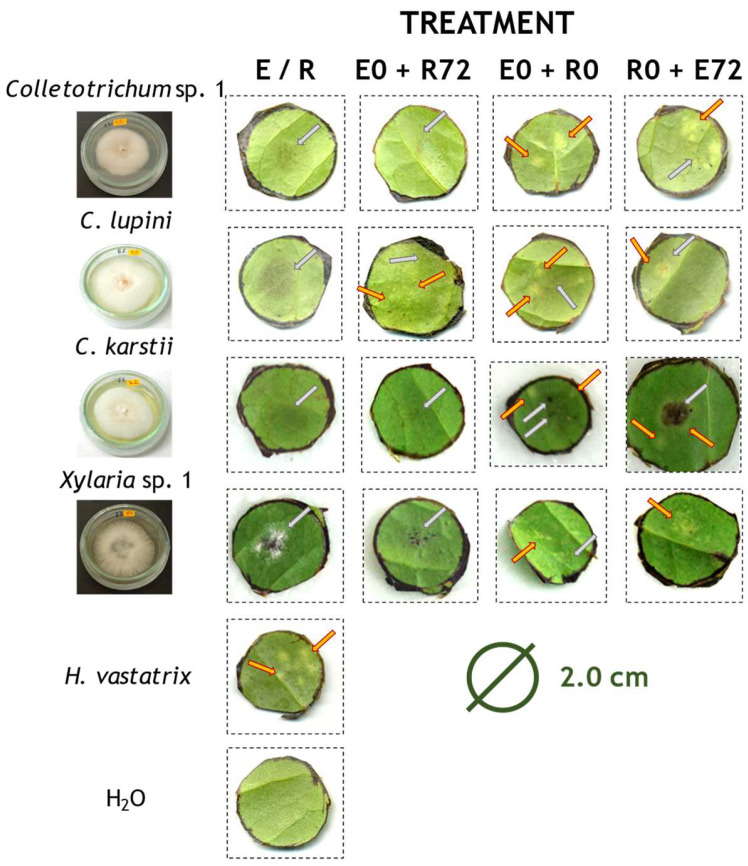
Example of Caturra coffee leaf discs 25 days after treatments inoculated with endophytes and CLR. E/R, endophyte or CLR (*n* = 20); E0 + R72, endophyte 72 h before inoculating CLR (*n* = 20); E0 + R0, both endophyte and CLR inoculated simultaneously (*n* = 20); R0 + E72, endophyte 72 h after CLR (*n* = 20); H_2_O, control. Images were corrected using GIMP 2.10 (www.gimp.org, accessed on 13 September 2022). Orange arrows highlight the presence of CLR and grey arrows the endophytes.

**Figure 2 plants-13-00814-f002:**
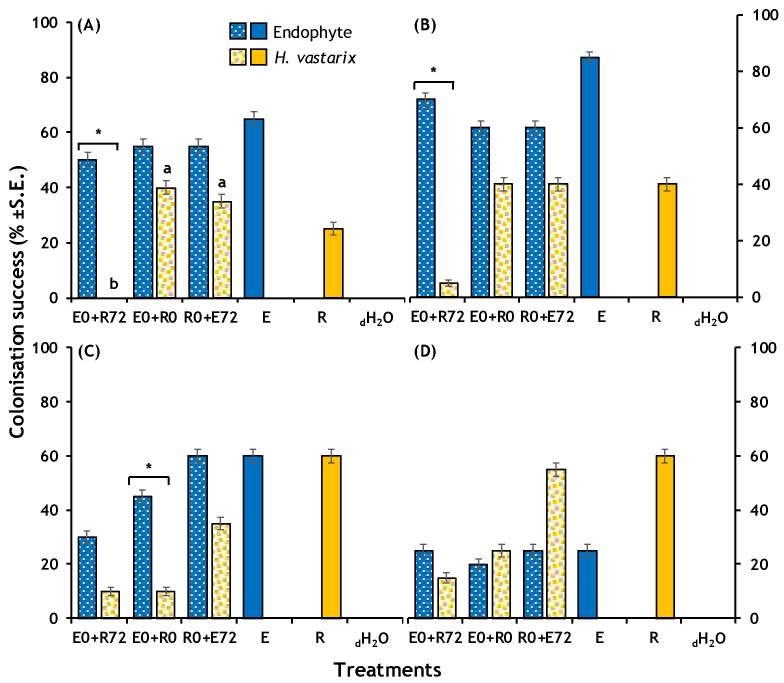
Colonisation success of four fungal endophytes and *Hemileia vastatrix* (CLR) of the leaf discs under different treatments. (**A**) *Colletotrichum* sp. 1 (F042); (**B**) *C. lupini* (F072); (**C**) *C. karstii* (F085); and (**D**) *Xylaria* sp. 1 (F131), compared to *H. vastatrix*. Treatments: E0 + R72, endophyte 72 h before inoculating CLR (*n* = 20); E0 + R0, both endophyte and CLR inoculated simultaneously (*n* = 20); R0 + E72, endophyte 72 h after CLR (*n* = 20). Controls: E, each endophyte (*n* = 20); R, *H. vastatrix* (*n* = 20); and _d_H_2_O, blank control (*n* = 20). The letters denote significant post-hoc test values based on adjusted standardised residuals for the colonisation of CLR across treatments. Colour filled bars are the control treatments and were not included in the analysis. Asterisks denote significant Chi-square test values for differences between the colonisation success of the endophyte and the CLR within treatments. There were no effects of treatment on CLR colonisation success when inoculating with *Xylaria* sp. 1.

**Figure 3 plants-13-00814-f003:**
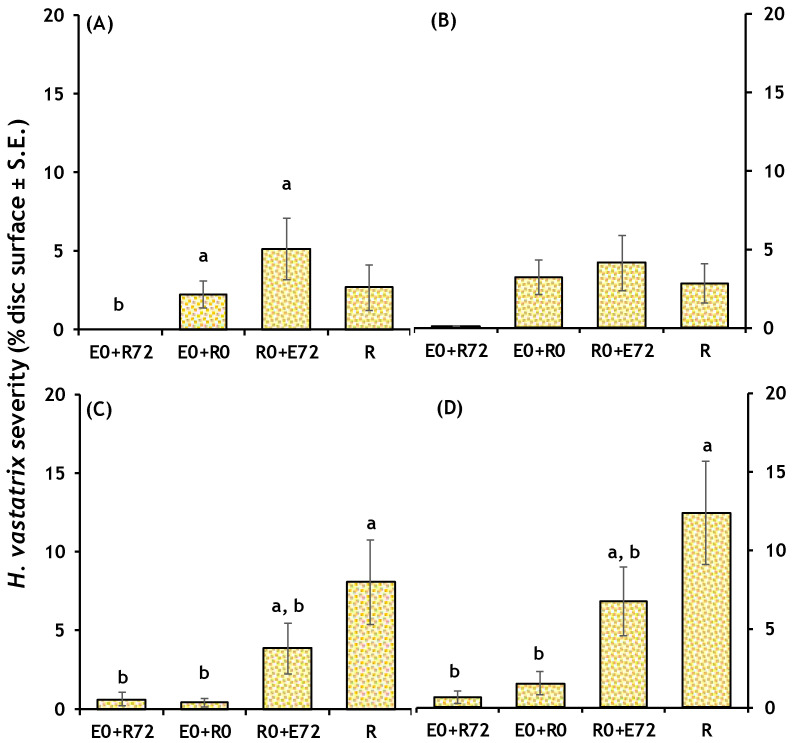
Leaf discs infection severity of *Hemileia vastatrix* when interacting with different endophyte species. (**A**) *Colletotrichum* sp. 1; (**B**) *C. lupini*; (**C**) *C. karstii*; and (**D**) *Xylaria* sp. 1. Treatments: E0 + R72, endophyte 72 h before inoculating CLR (*n* = 20); E0 + R0, both endophyte and CLR inoculated simultaneously (*n* = 20); R0 + E72, endophyte 72 h after CLR (*n* = 20). Controls: E, each endophyte (*n* = 20); R, *H. vastatrix*, CLR (*n* = 20); and _d_H_2_O, blank control (*n* = 20). Letters denote significant post-hoc pairwise comparisons corrected after Bonferroni.

**Table 1 plants-13-00814-t001:** Identification and relative frequency (RF) of fungal endophytes in Caturra and Colombia coffee varieties. The taxonomic group was assessed for the samples isolated in this work (NCBI GenBank IDs in brackets, otherwise determined after morphological resemblance) after blast similarity with the GenBank ID of the closest taxa (percent identity in bold). * Denotes samples that were identified in the pre-experiment, and were not included in the analyses; ** denotes sequences of low quality. Blue and light orange denote fungal species unique to each variety.

Taxonomic Group	Sample ID (GenBank ID)	Closest Taxa (GenBank ID: % ID)	RF (%) in Caturra	RF (%) in Colombia
Ascomycota				
Dothideomycetes				
Incertae sedis				
Fungal endophyte sp. 1	F112 (MK005037)	EU687041: **90.22**	0.00	12.50
Capnodiales; Cladosporiaceae				
Uncultured Cladosporiaceae	F129 **	GQ517141: **86.88**	12.50	0.00
Pleosporales; Sporormiaceae				
*Preussia pseudominima*	F041 (MK005019)	KU204603: **99.24**	0.00	12.50
*Preussia* sp.	F134 (MK005045)	KR093940: **95.20**	12.50	0.00
Eurotiomycetes				
Eurotiales; Aspergillaceae				
*Penicillium* sp.	F096, F126 (MK005040)	KF498874/MN788117/MN788106: **91.12**	12.50	12.50
Trichocomaceae				
*Talaromyces* sp.	F022 (MK005060), F081, F099	OM791640/MH934969/MT530189: **89.88**	12.50	25.00
Sordariomycetes				
Fungal endophyte sp. 2	F065 (MK005023)	KT289540: **96.19**	0.00	12.50
Diaporthales; Diaporthaceae				
*Diaporthe* sp.	F118 (MK005038)	MF280391/OM975589: **95.34**	12.50	0.00
Glomerellales; Glomerellaceae				
*Colletotrichum acutatum* complex				
*C. lupini*	F072 (MK005027), F073, F075, F082	MH178095: **99.14**	0.00	25.00
*C. scovillei*	F043 (MK017761), F045, F057	LC488868: **99.15**	0.00	12.50
*Colletotrichum* sp. 1	F042 (MK005049), F058	MK005027 (F072): 91.53 MK005048 (F040)/MH865411/MH854629: **91.45**	0.00	25.00
*Colletotrichum* sp. 2	F036, F039, F040 (MK005048), F046, F102	EF687919: **92.65**	0.00	50.00
*Colletotrichum* sp. 3	F127 (MK005055)	JQ894656: **91.74**	12.50	0.00
*Colletotrichum* sp. 4	F026, F132 (MK005044)	ON329227: **97.95**	25.00	0.00
*Colletotrichum* sp. 5	F113 (MK005058)	ON368204: **89.00**	0.00	12.50
*C. boninense* species complex				
*Colletotrichum* sp. 6	F066 (MK005051), F080 (MK005052)	MK005052 (F080): 97.70/MN458530: **97.30**/MT464454: 97.32	0.00	12.50
*Colletotrichum* sp. 7	F011 * (MK005046), F078, F084, F133	OL842171: **90.40**	12.50	37.50
*C. gloeosporioides* species complex				
*C. karstii*	F059 (MK005022), F083 (MK005031), F085 (MK005032), F050 (MK005020), F079 (MK005030), F077 (MK005029), F095 (MK005035), F031, F068, F089, F098, F100	OM436864: 96.95/MN842791: 97.66/OP445269: 98.14/KX578788: **98.80**/KX578788: 98.80/OM436864: 98.51/OP782678: 93.77/KX578788/MK005029: 93.52	0.00	87.50
*Colletotrichum* sp. 8	F053 (MK005050), F094	MK005055 (F127): **91.21**/OK030873: 89.36	0.00	25.00
*Colletotrichum* sp. 9	F007 * (MK005009), F029	OW988162: **94.84**/KX069828: 94.66	0.00	12.50
Glomerellales; Plectosphaerellaceae				
*Musicillium* sp.	F017 *, F019 * (MK005015), F103	MK579179: **95.84**	0.00	12.50
Hypocreales; Clavicipitaceae				
*Beauveria* sp.	F117		12.50	0.00
Sordariales				
Fungal endophyte sp. 3	F104 **	KF435260: **86.97**	0.00	12.50
Sordariales; Chaetomiaceae				
*Chaetomium* sp.	F060, F074 (MK005028)	MF495440/KF435950/KF435726/KF435552/KF435385: **87.70**	0.00	25.00
Trichosphaeriales; Trichosphaeriaceae				
*Nigrospora* sp. 1	F025 (MK005047), F030, F034 **	MT123068: **91.94**	12.50	25.00
*Nigrospora* sp. 2	F128 (MK005041)	MN341467: **89.74**	12.50	0.00
Xylariales; Hypoxylaceae				
*Annulohypoxylon* cf. *stygium*	F015 (MK005013)	KP133169: 98.75	12.50	0.00
*Hypoxylon* sp. 1	F071 (MK005026)	FJ612775: **96.97**	0.00	12.50
Xylariales; Xylariaceaea				
Fungal endophyte sp. 4	F106 (MK005036)	EU687119: **97.13**	0.00	12.50
Fungal endophyte sp. 5	F064 **	KU747690/FJ612923: **75.39**	0.00	12.50
*Anthostomella* sp.	F121 (MK005059)	JQ754021: **91.14**	12.50	0.00
*Lopadostoma* sp.	F001 *, F002 * (MK005007)	KC774600: **91.11**	12.50	0.00
*Nemania* sp.	F125 (MK005039)	KF435731: **97.88**; MF770851: 97.38	12.50	0.00
*Xylaria* sp. 1	F086 (MK005033), F035 (MK005017), F032 (MK005016), F087 (MK005034), F097 (MK005054), F131 (MK005043), F047, F048, F055	KP133288: **96.92**, 95.79, 95.07, 95.60, 93.68, and 95.18	12.50	87.50
*Xylaria* sp. 2	F008 * (MK005010), F021	KF467102: **94.59**	12.500	0.00
*Xylaria* sp. 3	F016 * (MK005014), F114	MH003490: **96.34**	0.00	12.50
*Xylaria* sp. 4	F067 (MK005024), F037 (MK005018), F010 *, F063	KF435704: **98.54**,/MK005018 (F037) 97.24	0.00	25.00
*Xylaria* sp. 5	F023 (MK005056)	MH003401: **92.02**	12.50	0.00
*Xylaria* sp. 6	F130 (MK005042)	MT992054: **94.33**	12.50	0.00
*Xylaria* sp. 7	F027 **, F044, F056 (MK005021), F061, F062, F076, F101	JQ341084: **98.23**	0.00	87.50
*Xylaria* sp. 8	F088 (MK005053), F012 *, F051, F054, F124	MK334005/MK247857: **98.75**	12.50	37.50
*Xylaria* sp. 9	F070 (MK005025)	KJ883611: **97.21**	0.00	12.50
*Xylaria* sp. 10	F009 * (MK005011), F052	MN833802/KT289626/KP13343: **94.22**	0.00	12.50
Basidiomycota				
Agaricomycetes				
Hymenochaetales; Schizoporaceae				
*Xylodon* sp.	F107 **	OM891735: **87.3**	0.00	12.50
44 taxonomic groups			20 spp.	30 spp.

**Table 2 plants-13-00814-t002:** Endophyte diversity indexes, evenness, and richness in two coffee varieties.

	Caturra (± C.I.)	Colombia (± C.I.)
Taxa	20	30
Individuals	21	62
Simpson’s *D*	0.005 ± 0.029	0.043 ± 0.013
Unbiased Shannon’s *H_CS_*	7.955 ± 1.120	3.491 ± 0.100
Margalef’s *K*	6.241 ± 0.655	7.027 ± 0.127
Whitakker’s *β*-diversity	0.76	
*β* _sim_	0.70	

**Table 3 plants-13-00814-t003:** Pearson’s Chi-square test for the effects of the treatments (E + R72, E + R0 and R + E72) on the colonisation success of endophytes and CLR. *N* = 120. *p*-Values in bold denote statistically significant effects of treatment on colonisation success.

	Pearson’s *χ*^2^	d.f.	*p*-Value
*Colletotrichum* sp. 1 (F042)	0.134	2	0.935
CLR	10.133	2	0.006
*C. lupini* (F072)	0.574	2	0.750
CLR	8.044	2	0.018
*C. karstii* (F085)	3.636	2	0.162
CLR	5.566	2	0.062
*Xylaria* sp. 1 (F131)	0.186	2	0.911
CLR	8.010	2	0.018

**Table 4 plants-13-00814-t004:** The Kruskal-Wallis test for the effects of endophyte identity and the treatment on CLR severity in infecting leaf discs. Analysis of the main effects and the specific interaction of each endophyte. Values in bold denote significant *p*-values.

	N	*H*	d.f.	*p*-Value
Treatment	320	36.790	3	**<0.001**
*Colletotrichum* sp. 1 × Treatment	80	9.531	3	**0.023**
*C. lupini* × Treatment	80	8.969	3	**0.030**
*C. karstii* × Treatment	80	17.330	3	**0.001**
*Xylaria* sp. 1 × Treatment	80	16.121	3	**0.001**

## Data Availability

The occurrence of endophytes and nucleotide data (GenBank IDs) are summarised in Table 1. Data on colonisation success and severity will be deposited in Dryad repository after acceptance.

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
