# Peer review of "Diversity of Leaf Fungal Endophytes from Two Coffea arabica Varieties and Antagonism towards Coffee Leaf Rust"

_plants, 2024, doi:10.3390/plants13060814_

Round 1
Reviewer 1 Report
Comments and Suggestions for Authors
Diversity of leaf fungal endophytes from two Coffea arabica varieties and antagonism towards Coffee Leaf Rust
By Ruth A. Poma-Angamarca et al.
The author list is odd:
Ruth A. Poma-Angamarca 1 , Jacqueline R. Rojas 1 , Aminael Sanchez-Rodriguez 1, §, Mario X. Ruiz-González 1, 2, 3, § and *
Why have ‘and *’ there?!
The manuscript is certainly of interest to scientists working on endophytes, particularly in coffee. Although quite well written, in places the use of English is rather poor: the authors must seek the help of a native speaker of English, preferably someone who has some knowledge of plant pathology/mycology.
First sentence of the abstract – ‘Coffee is a valuable commodity appreciated worldwide.’ – massively underplays the role of coffee as a global commodity: its value is immense.
‘Sp.’ and ‘spp.’ must NOT be in italics.
Line 4 of introduction: close the bracket on ‘(e.g., [1]’
Page 1, Introduction: ‘Under up-to-date climatic predictions, we expect a drastic reduction of 19 – 50 % of the suitable areas for coffee production’ – does this mean areas within currently productive places? Is it possible that other areas might become suitable for production though?
Page 2, Introduction: CLR is caused by the obligate biotrophic basidiomycete fungus Hemileia vastatrix ‘ use up-to-date taxonomy (Pucciniomycetes, or Pucciniales).
Page 2, introduction: ‘Colletotrichum gloeosporoides o …’ – presumably it should be ‘or’?
Results
Page 3, results: ‘between three to twelve days.’ Between three to twelve days since what?
Page 3, results: ‘closest taxids to 31 of the isolated fungal’ – I think they mean ‘taxa’.
Page 3, results: ‘eight as diseases’ – the fungi themselves are not diseases. The disease is the problem they cause. They are pathogens.
Page 3, results: ‘frequency of 87.50 %. X. adscendens’ – do not use the abbreviated genus name when it is the first word in a sentence.
Page 3, results, Table 1 caption: ‘Days’’ – just say ‘the Days column. Do not use the possessive (‘) as it is completely unnecessary.
Page 7, results, footnote to Table 1: Why are the references given in the footnote? They could perfectly logically be in the full reference list at the end of the paper.
Page 7, results: ‘The bias-corrected Shannon entropy’s Hcs value in Caturra…’ get ride of the ‘ in entropy!
Page 7 onwards: 2.2.1. Colonisation success – please write in the past tense.
Pages 7 – 8: ‘Overall, there is a trend that suggests that the presence of the endophyte, when previously inoculated, reduced CLR colonisation success’ – was the trend significant? You do not say that it was. If there was no significant difference, then the trend may not be real. You need to change this statement to include the significance described in the next two sentences.
Page 8, Figure 1: Caption – ‘Example of Caturra coffee leaf discs after 25 days treatments inoculated with endophytes and CLR’. Change to ‘…25 days after treatments inoculated…’ The symptoms of leaf rust are not very clear at all.
Page 9, Figure 2: Caption needs something to explain what the ‘R’ treatment was. The legend on the figure also fails to identify R. I assume it is rust-inoculated alone, but it must be specified.
Page 9, CLR severity in leaf discs infection: ‘The Kruskal-Wallis test found overall significant effects for treatment…’ – Significant effects for treatment… were found when using the Kruskal-Wallis test…’
The discussion is fine in a limited manner, but not adequate for the results presented. Is there a possibility that some of the identified endophytes may become pathogenic under certain circumstances (for example). An important point to consider is what proportion of the true endophytic fungal community was obtained using the fairly basic isolation methods applied here? Moreover, we can assume that the leaves used in the leaf disc assays also had endophyte communities: it is difficult to control for that possibility, but it is worthy of some discussion.
Page 11, Methods: ‘We sampled three phenotypically healthy leaves from eight plants…’ – what was the phenological age of the leaves? The endophyte communities are likely to change with leaf age.
Pages 11 – 12: How were the leaves stored prior to isolation work?
Page 12: Did you only re-isolate the fungi that grew out of tissues by day 3 of incubation? I am sure there would be other fungi that took longer than 3 days to emerge from the host tissues.
Page 12, Methods: ‘Endophyte biodiversity’ section – do not use ‘didn’t’ – it is slang. Did not.
Page 12, Methods: 4.3. Cross-inoculation essays in leaf discs – should be ‘assays’, not essays.
Page 12, Methods: ‘one Xilaria adscendens endophyte,…’ correct the spelling of Xylaria.
Page 13, Methods: ‘incubated with a 12 hours photoperiod’ – how was the ‘photoperiod’ produced? What light intensity was used?
Page 13, Methods: ‘25 days post inoculation leaf discs…’ – do not use numerals for the first word in a sentence. ‘Twenty-five days post-inoculation…’
Comments on the Quality of English LanguageSee above. I have included comments on the language in the main review.
Author Response
We thank you very much for your review. The manuscript has gained clarity and strength, and we hope to have satisfied all your observations and doubts.
Reviewer 1
The author list is odd: Ruth A. Poma-Angamarca 1 , Jacqueline R. Rojas 1 , Aminael Sanchez-Rodriguez 1, §, Mario X. Ruiz-González 1, 2, 3, § and *
Why have ‘and *’ there?!
- We have fixed the typo, thank you very much.
The manuscript is certainly of interest to scientists working on endophytes, particularly in coffee. Although quite well written, in places the use of English is rather poor: the authors must seek the help of a native speaker of English, preferably someone who has some knowledge of plant pathology/mycology.
- We followed the reviewer’s advice and the text has been strongly improved.
First sentence of the abstract – ‘Coffee is a valuable commodity appreciated worldwide.’ – massively underplays the role of coffee as a global commodity: its value is immense.
- The reviewer is right, so now we say “Coffee has immense value as a worldwide appreciated commodity”
‘Sp.’ and ‘spp.’ must NOT be in italics.
- This has been fixed.
Line 4 of introduction: close the bracket on ‘(e.g., [1]’
- This has been fixed
Page 1, Introduction: ‘Under up-to-date climatic predictions, we expect a drastic reduction of 19 – 50 % of the suitable areas for coffee production’ – does this mean areas within currently productive places? Is it possible that other areas might become suitable for production though?
Page 2, Introduction: CLR is caused by the obligate biotrophic basidiomycete fungus Hemileia vastatrix ‘ use up-to-date taxonomy (Pucciniomycetes, or Pucciniales).
- Now we indicate the taxonomic ranks for the species: “(class Pucciniomycetes, order Pucciniales, family Zaghouaniaceae)”
Page 2, introduction: ‘Colletotrichum gloeosporoides o …’ – presumably it should be ‘or’?
- You are right. We fix it.
Results
Page 3, results: ‘between three to twelve days.’ Between three to twelve days since what?
- We wrote a clearer sentence: “The endophytic fungi took between three and twelve days to grow from the plant tissue after sowing the leaf fragments.”
Page 3, results: ‘closest taxids to 31 of the isolated fungal’ – I think they mean ‘taxa’.
- That is fixed.
Page 3, results: ‘eight as diseases’ – the fungi themselves are not diseases. The disease is the problem they cause. They are pathogens.
- We apologise for such a basic mistake.
Page 3, results: ‘frequency of 87.50 %. X. adscendens’ – do not use the abbreviated genus name when it is the first word in a sentence.
- That was fixed
Page 3, results, Table 1 caption: ‘Days’’ – just say ‘the Days column. Do not use the possessive (‘) as it is completely unnecessary.
- Done
Page 7, results, footnote to Table 1: Why are the references given in the footnote? They could perfectly logically be in the full reference list at the end of the paper.
- We agree with the reviewer. However, after Reviewer 2 comments, we have moved part of the table to supplementary material, and so the references did.
Page 7, results: ‘The bias-corrected Shannon entropy’s Hcs value in Caturra…’ get ride of the ‘ in entropy!
- We fixed that
Page 7 onwards: 2.2.1. Colonisation success – please write in the past tense.
- We have corrected the verbal tenses.
Pages 7 – 8: ‘Overall, there is a trend that suggests that the presence of the endophyte, when previously inoculated, reduced CLR colonisation success’ – was the trend significant? You do not say that it was. If there was no significant difference, then the trend may not be real. You need to change this statement to include the significance described in the next two sentences.
- Thanks for highlighting this. We misused the word “trend” because we meant a tendency in the sense of the observation of an inclination, not of a statistical confirmation. Therefore, now we say: “Overall, it is observed that when the endophyte has been previously inoculated, the CLR colonisation success is reduced (Fig. 2). The latter was confirmed when analysing for differences within each treatment for…”
Page 8, Figure 1: Caption – ‘Example of Caturra coffee leaf discs after 25 days treatments inoculated with endophytes and CLR’. Change to ‘…25 days after treatments inoculated…’
- We corrected the caption.
The symptoms of leaf rust are not very clear at all.
- The reviewer is right, so we have sought advice about how to improve the graphical results of leaf disc symptoms. We found some better images, corrected the pictures’ brightness and contrast with the GIMP 2.10, outlined pictures for better clarity and, after printing the figure in colour, we manage to distinguish the pale symptoms of CLR and the dark areas where endophyte mycelium is growing. We added a line in Acknowledgements: “; A. KabaÅŸ for advice on how to improve the graphical results of leaf discs symptoms”
Page 9, Figure 2: Caption needs something to explain what the ‘R’ treatment was. The legend on the figure also fails to identify R. I assume it is rust-inoculated alone, but it must be specified.
- The legend states “R, CLR (n = 20)”, However, for more clarity, we changed to “R, H. vastatrix (n = 20);”
Page 9, CLR severity in leaf discs infection: ‘The Kruskal-Wallis test found overall significant effects for treatment…’ – Significant effects for treatment… were found when using the Kruskal-Wallis test…’
- We changed the sentence
The discussion is fine in a limited manner, but not adequate for the results presented. Is there a possibility that some of the identified endophytes may become pathogenic under certain circumstances (for example).
- The reviewer is right. Now have added: “The output of biotic interactions, however, is complex, and changes in the environment might drive changes in the lifestyle, from mutualist to saprophyte or pathogenic [53].”
An important point to consider is what proportion of the true endophytic fungal community was obtained using the fairly basic isolation methods applied here?
- Indeed, that is a very good question, that we could only answer by combining both our fairly basic isolation method and metagenomic analysis of the leaves. We will keep it in mind for future works.
Moreover, we can assume that the leaves used in the leaf disc assays also had endophyte communities: it is difficult to control for that possibility, but it is worthy of some discussion.
- Now we say: “However, although we used 10-month-old young plants that were raised in a greenhouse, it is possible that their leaves were already colonised with unidentified endophytes, and that could have contributed to ameliorating CLR severity. Nevertheless, our results are exploratory and more experiments should be conducted on living plants instead of on leaf discs.”
Page 11, Methods: ‘We sampled three phenotypically healthy leaves from eight plants…’ – what was the phenological age of the leaves? The endophyte communities are likely to change with leaf age.
- Coffee plants are evergreen and leaves can remain in the plant all year round. However, under field conditions the leaves get covered with sooty molds, lichens, mosses, scratches, insect bites, etc. The older the leaf, the larger the collection of epiphytes and injuries. Thus, by choosing phenotypically healthy leaves we are selecting for both similar age leaves and leaves with a healthy endophyte community because the leaves are less prone to host environmental infections acquired through injuries.
Pages 11 – 12: How were the leaves stored prior to isolation work?
- Leaves were stored in aseptic paper bags and transported to the laboratory within the hour.
Page 12: Did you only re-isolate the fungi that grew out of tissues by day 3 of incubation? I am sure there would be other fungi that took longer than 3 days to emerge from the host tissues.
- We checked the leaf fragments for two months. So we wrote a clearer sentence: “We only considered as endophytes those fungi that grew after a minimum of three days of culturing,…”
Page 12, Methods: ‘Endophyte biodiversity’ section – do not use ‘didn’t’ – it is slang. Did not.
- We fixed it.
Page 12, Methods: 4.3. Cross-inoculation essays in leaf discs – should be ‘assays’, not essays.
- That was a typo, thank you very much.
Page 12, Methods: ‘one Xilaria adscendens endophyte,…’ correct the spelling of Xylaria.
- We fixed it.
Page 13, Methods: ‘incubated with a 12 hours photoperiod’ – how was the ‘photoperiod’ produced? What light intensity was used?
- Actually, when talking about the 12 hours photoperiod we refer to the hours of natural light that enters through the laboratory windows, since at our latitude (3.9952° S) the day has approximately 12 hours of sunlight.
Page 13, Methods: ‘25 days post inoculation leaf discs…’ – do not use numerals for the first word in a sentence. ‘Twenty-five days post-inoculation…’
- We corrected it.

Reviewer 2 Report
Comments and Suggestions for Authors
Dear Authors,
The MS “Diversity of leaf fungal endophytes from two Coffea arabica varieties and antagonism towards Coffee Leaf Rust” reports some useful data; however, I have some comments and suggestions to improve the MS.
Best wishes

Author Response
We thank you very much for your review. The manuscript has gained clarity and strength, and we hope to have satisfied all your observations and doubts.
Reviewer 2
Recheck all. sp. and not in italics and also spp.
sp.
spp. and not in italics
- We have checked all “sp.” “spp.”, including figure 1.
Keywords should not replete with the title
- We have removed two redundant keywords with the title.
Please add how importance of coffee in your study region and also including economic value
- Now we added some information: “Ecuador produces around 8000 t of coffee in 30000 ha (SIPA, http://sipa.agricultura.gob.ec/), which represents an annual income of about USD 150 mil-lion (http://www.camae.org/).”
use em dash between the number
- Fixed
Add family, order
- Now we indicate a more complete taxonomic description: “(class Pucciniomycetes, order Pucciniales, family Zaghouaniaceae).”
Add endophytic fungi associated with coffee and their benefits. Example: -Lu, L., Karunarathna, S.C., Hyde, K.D., Suwannarach, N., Elgorban, A.M., Stephenson, S.L., Al-Rejaie, S., Jayawardena, R.S. and Tibpromma, S., 2022. Endophytic fungi associated with coffee leaves in China exhibited in vitro antagonism against fungal and bacterial pathogens. Journal of Fungi, 8(7), p.698.
- That is a very nice piece of work and we apologise for not including it in the previous version. Now we say: “Previous works on coffee found bacterial endophytes acting as plant growth promoters, or potential agents to control bacterial or fungal diseases, such as CLR [29-31].
which varieties?
please mention, country/city or other
- We made the sentence clearer: “In this work, we investigated the fungal endophyte community associated with two common C. arabica varieties, Caturra and Colombia, from Southern Ecuador (Shucos, Loja province).”
Which species?
any references?
please check more publications this genus is also well known that can produce several secondary metabolites
- Now we say: “two Colletotrichum species that are known pathogens of different host plant species, C. lupini and C. karstii [42, 43], an undescribed species from the Colletotrichum acutatum species complex, and a fungus from a genus known both to prosper on decaying plant material and produce a wide variety of bioactive secondary metabolites, Xylaria adscendens [44,45].”
please add why only fungal were selected to do leaf discs test
- We imagine that you ask why we did not study bacterial diversity and their effects on leaf discs. Well, first, this work was designed as a Master’s project, and both the student’s time of dedication and the budget for the project were decisive in choosing more or less clear and achievable objectives. However, we have ongoing research on Coffea arabica, and we keep in mind working with bacterial diversity as well. We are not sure if you still want us to add this to material and methods section.
taxa?
- Fixed
the table can move to supplementary
- After giving many thoughts to the reviewer's comment, we partially agree that parts of the table 1 could go to supplementary material (closest taxa from GenBank and their ecology). However, we believe it is important to present the reader with a list of endophyte species and their IDs in GenBank. On the other hand, the percentage of identity with the GenBank sequences shows that there are many potentially new taxa. Thus, we have moved to supplementary material the
consistent with the format
- We fixed that issue.
The author should mention the practical applicability of these endophytic fungi and the challenges while using these fungi and also why a special focus on endophytic isolates?
- Now we say:
“Nevertheless, our results are exploratory and more experiments should be conducted on living plants instead of on leaf discs, before determining the practical applicability of these endophytic fungi.”
“In addition, although a Colletotrichum sp. strain has been found to improve growth and plant secondary metabolism [52], the genus hosts many pathogenic fungi, and, thus, the inoculation of plants with these species must be carefully monitored under different environmental situations and host genetic backgrounds, as well as to avoid breakthrough infections in other crop hosts. Xylariaceae members, however, produce a great diversity of secondary metabolites, and some species even show antagonistic effects against pathogens [44, 53], thus, representing a source of unknown bioactive components with potential application in agronomy, industry, medicine and biotechnology. Indeed, focusing on studying endophyte isolates not only contributes to characterizing biodiversity and microbial specificity, but also represents the opportunity to bio-prospect new compounds with great potential for the development of green technologies and promoting sustainable practices”
Did you have a control group for the surface sterilized method, whether your sterilization is correctly done?
- No, we did not need that kind of control, because when the surface of a fragment is not properly sterilised, fungus and bacteria grow from the surface of the fragment within 24 h, and thus can be easily removed.
simple text
- We fixed.
Please specific
- 20 ºC
are you sure?
- Yes, the plates with leaf fragments were kept for 60 days, and were checked nearly every day for fungi emergence.
Faster growers might be epiphytes or, eventually, a contamination.
- We removed the sentence
please add the accessed date as these results can be changed when more sequences are deposited in GB
- We now say: “(December 2023)”

Round 2
Reviewer 2 Report
Comments and Suggestions for Authors
Dear authors,
I have a few suggestions in the attached file. Please carefully check and improve.
Thanks

Author Response
Answers to referee 2 comments:
R: so why is different as in the end of introduced mentioned "The former is susceptible, and the latter is resistant to CLR"
A: We are not sure about what you are asking for specifically. However, we understand that some clarification is needed. Now, we removed that sentence and say at the end of the introduction: “The former, being susceptible, and the latter, being resistant to CLR, provide an opportunity to explore the potential interaction between host genotype and fungal endophyte selection in the field.”
R: We isolated 44 morphotypes and found high endophyte diversity in both varieties, which are associated to two statistically different fungal communities.
A: We deleted the sentence from the abstract
R: arrange in alphabetical order
A: it has been done.
R: Pucciniomycetes, Pucciniales, Zaghouaniaceae
- We have done the change
R: I strongly suggest mentioning only genus name (e.g., Colletotrichum sp. 1, Colletotrichum sp. 2 .......) as this is only ITS gene Blast search and later, many sequences will submit to GB
A: We agree with the reviewer. Only in the few cases where the ID was above 98.5 %, we left the complete name of the species. We have changed appropriate changes across text, tables, and figures when necessary.
R: Change all to "colonization "
A: Please note that we are utilizing British English for the text. In British English, the correct spelling of the word is 'colonisation,' whereas in American English, it is spelled 'colonization.' Thank you for your attention to this detail. We found one misspelling and corrected it.
R: Can you add control photo as well
A: Certainly. We added the picture of the leaf disc.
R: How do you confirm it's your endophytes or not? Did you re-isolate it?
A: Our methodology ensured sterile conditions throughout the experiment. The control leaf discs remained free of infection, validating our procedures. Then, the morphology of the mycelia observed in the treated discs closely matched that of the inoculated endophyte species, confirming their presence. We did not re-isolate the fungi.
R: in the picture used E0, recheck all
A: Thank you very much for your observation. We have fixed it.
R: I would like you to add work that needs to be done in the future, like identifying fungal species or doing another experiment
A: That is a nice suggestion. We have added “The endophytes from the Colombia variety successfully grew in Caturra variety leaf discs, indicating their adaptability across different host genotypes. However, it appears that there is a fungal preference for specific host genotypes in the field. New experiments should first investigate the potential pathogenicity of these endophytes in whole coffee plants before exploring their protective role against different CLR strains.”
I attached the manuscript with tracking changes. Please see the attachment.
